# Anxiety and depression symptoms, the recovery from symptoms, and loneliness before and after the COVID-19 outbreak among the general population: Findings from a Dutch population-based longitudinal study

**Peter G. van der Velden**[1,2]*, **Philip Hyland**[3], **Carlo Contino**[4], **Hans-Martin von Gaudecker**[5], **Ruud Muffels**[6], **Marcel Das**[1,7]

1 CentERdata, Tilburg, The Netherlands, 2 Tilburg University's Network on Health and Labor (NETHLAB), Tilburg, The Netherlands, 3 Department of Psychology, Maynooth University, Kildare, Ireland, 4 Fonds Slachtofferhulp, The Hague, The Netherlands, 5 Institute for Applied Microeconomics, Universität Bonn, Bonn, Germany, 6 Tilburg School of Social and Behavioral Sciences, and Tranzo (Scientific Center for Care and Welfare), Tilburg University, Tilburg, the Netherlands, 7 Tilburg School of Economics and Management, Tilburg University, Tilburg, The Netherlands

* pg.vandervelden@tilburguniversity.edu

## Abstract

### Objectives

Examine the effects of the COVID-19 pandemic on the mental health and loneliness in the general population. More specifically, the study focused on prevalence of anxiety and depression symptoms, the extent to which individuals with existing symptoms recovered or not, the prevalence of subtypes of loneliness, and the extent to which loneliness before and during this pandemic was associated with anxiety and depression symptoms.

### Methods

Data was extracted from the longitudinal LISS panel, based on a probability sample of the Dutch population, with assessments on loneliness in October 2019 (T1) and June 2020 (T4), and anxiety and depression symptoms in November 2019 (T2), March 2020 (T3) and June 2020 (T4; $N^{total}$ = 4,084). Loneliness was examined with the De Jong Gierveld Loneliness Scale and anxiety and depression symptoms with the Mental Health Inventory (MHI-5).

### Results

Repeated measures multivariate logistic regression analyses (RMMLRA) showed a statistical significant lower prevalence of anxiety and depression symptoms after the outbreak (T4 = 15.3%) than before (T2 = 16.8%) and during the COVID-19 outbreak (T3 = 17.2%). According to the Reliable Change Index, the distribution of recovery categories (remission, improvement, unchanged and worsening symptoms) after the outbreak did not differ

**Data Availability Statement:** The study was conducted using the Dutch Longitudinal Internet studies for the Social Sciences (LISS) panel [27]. The LISS panel started in 2007 and is based on a large traditional probability sample drawn from the Dutch population. The Dutch Research Counsel (NWO) funded the set-up of LISS. Panel members receive an incentive of €15 per hour for their participation and those who do not have a computer and/or Internet access are provided with the necessary equipment at home. Further information about all conducted surveys and regulations for free access to the data, as well as the data files can be found at www.lissdata.nl (in English). The LISS panel has received the international Data Seal of Approval (see https://www.datasealofapproval.org/en/). All data of studies conducted with the LISS panel are anonymized.

**Funding:** This work was supported by Fonds Slachtofferhulp, The Haque, The Netherlands (50006/VICTIMS). One of the authors (CC) is deputy director at Fonds Slachtofferhulp. The funder had no further role in study design, data collection and analysis, decision to publish, or preparation of the manuscript.

**Competing interests:** The authors have declared that no competing interests exist.

significantly from the distribution of these categories before the outbreak. RMMLRA revealed that the prevalence of emotional loneliness increased significantly after the outbreak (T1 = 18.4%, T4 = 24.8%). Among individuals who were not lonely before and after the outbreak the prevalence of symptoms decreased significantly (T2 = 7.0%, T4 = 4.4%) and, likewise, among those who were not lonely anymore after the outbreak (T2 = 21.5%, T4 = 14.5%). However, the prevalence of symptoms increased significantly among those who became lonely during the pandemic (T2 = 17.9%, T4 = 26.3%).

## Conclusions

Findings suggest that this pandemic did not negatively affect the prevalence of anxiety and depression symptoms nor the normal recovery of symptoms among the general population during the first four months, but that emotional loneliness increased.

## Introduction

The COVID-19 pandemic [1] negatively affects countries on a micro-, meso- and macro-level. At the micro-level it varies from the threat of people becoming infected or when infected, becoming very ill or even die, loss of significant others, unemployment, and social isolation because of social distancing rules, to reduced health care use among non-COVID-19 patients [2]. At meso and macro-levels it leads to understaffing and reduced revenues of firms, increasing governmental deficits, increasing economic insecurity, changing demand patterns of consumers [3], and collapsing growth levels in the global economy [4, 5]. A key question is to what extent this pandemic and the uncertainties about the future will negatively affect the mental health of the general population.

Several studies suggest that this pandemic increases the risk of mental health problems among the general population [6, 7]. However, many of these studies used cross-sectional study designs and are based on convenience samples instead of probability samples of the general population. A recent review by Xiong et al. [7] showed that all 19 identified and included COVID-19 studies used cross-sectional study designs and that 18 out of 19 studies used (potentially biased) convenience samples. These convenience samples used, for instance, snowball techniques and messages on social media as strategies to recruit respondents.

### Mental health problems

To the best of our knowledge, to date five prospective population-based studies with pre- and post-COVID-19 data have been published using probability samples of the general population to assess the effects of the COVID-19 pandemic on the mental health of the general population. Twenge and Joiner [8] assessed the effects in the United States (US) by comparing anxiety and depression symptomatology (according to the PHQ-2 and GAD-2) of adults participating in the National Health Interview Survey (NHIS; January-June 2019) with adults participating in the Household Pulse Survey (HPS; April-May 2020). They found that American adults participating in the HPS were three to four times as likely to screen positive for anxiety or depression disorders, or screen positive for both in April–May 2020, compared to American adults who were assessed in the first half of 2019. During the pandemic about 30% screened positive for these disorders. Importantly, the NHIS participants were asked about symptoms in the last two weeks and HPS participants were asked about symptoms in the last seven days. The study

by McGinty et al. [9] found that of the adult Americans who participated in the Johns Hopkins COVID-19 Civic Life and Public Health Survey (April 2020) 13.6% reported symptoms of serious psychological distress compared to 3.9% of the adults who participated in the National Health Interview Survey (NHIS) in 2018 according to the Kessler 6 scale. Daly et al. [10] compared the prevalence of depression symptoms according to the PHQ-2 among American adults participating in the National Health and Nutrition Examination Survey (NHNE) of 2017–2018, and among American adults who participated in the Understanding America Study (UAS) in March and April 2020. Results showed a significant higher prevalence in March (10.6%) and April (14.4%) 2020 UAS samples compared to the 2017–2018 (8.7%) NHNE sample. Pierce et al. [11] focused on UK residents of 16 years and older who participated in the UK Household Longitudinal Study and were asked to complete a COVID-19 web survey. Their findings showed that mean scores on the GHQ-12 increased significantly from 11.5 in 2018–2019 to 12.6 in April 2020. The increase was not simply a continuation of previous upwards trends from 2014 to 2019. The prevalence of clinically significant mental distress according to the GHQ-12, which should be distinguished from a mental disorder, among the total study sample increased from 18.9% in 2018–2019 to 27.3% in April 2020. The study by Van der Velden et al. [12] focused on Dutch adults using the data of surveys conducted with the population-based longitudinal LISS panel. Contrary to the previous population-based studies, no increase was found in anxiety and depression symptoms according to the MHI-5 during the COVID-19 outbreak (March 2020) compared to pre-outbreak levels (November 2019) among the general population. To account for possible seasonal effects, they also assessed the course of symptoms in the same period one year earlier (November 2018- March 2019), showing a similar pattern ruling out the possibility that findings can be attributed to a seasonal effect.

The differences in results between the studies in US and UK on the one side and in the Netherlands on the other side may partly be a result of (slightly) different study periods, besides differences in instruments, use of different samples to compare pre- and post-outbreak prevalence [8–10], governmental responses, and health care and social welfare systems. Perhaps the negative effects of the pandemic on the mental health of the general population are especially prevalent in the months after the COVID-19 outbreak.

## Recovery of symptoms

The sudden societal changes and related stressors after the COVID-19 outbreak may also have worsened symptoms or have hindered the recovery of symptoms among individuals with existing mental health problems. In addition, the pandemic may have reduced access to or use of mental health care services, or forced mental health care workers to continue the treatment using online facilities instead of usual vis-à-vis contacts contributing to a deteriorating mental health [13, 14].

However, the review by Sheridan Rains et al. [14] revealed that most studies on this important topic documented observations and experiences rather than reporting on research based on empirical data. To the best of our knowledge, to date only one empirical study focused on this issue. Based on a cross-sectional case-control study conducted at the end of February 2020, Hao et al. [15] revealed that non-psychotic psychiatric patients had higher COVID-19 related PTSD symptoms, and higher anxiety, depression, and stress symptom levels than the healthy control group. However, it is unknown if differences in anxiety, depression, and stress symptom levels were larger than under 'normal' circumstances. No empirical study has been conducted comparing pre-outbreak recovery-rates with post-outbreak recovery rates among those with existing mental health problems.

## Loneliness

Lockdowns and social distancing (including stay-at-home orders) are important preventive measures that may help to reduce the spread of COVID-19, but social isolation could lead to a rise of loneliness [16, 17] creating the so-called social connectivity paradox [16]. Non-COVID-19 studies have shown that loneliness, e.g. the subjective feeling referring to the perceived inadequacy of one's social relationships, is very prevalent and may be considered a major public concern. Loneliness is negatively associated with all-cause mortality, worse mental health and negative cardiovascular outcomes [18–20].

The limited number of COVID-19 related longitudinal studies on loneliness among the general population showed somewhat mixed results. In the earlier mentioned study by McGinty et al. [9], 13.8% always or often felt lonely in April 2020. Based on a comparison with another national survey conducted in April-May 2018, the authors concluded that loneliness increased only slightly (from 11% to 13.8%). Bu et al. [21] compared loneliness data of the large UK Household Longitudinal Study (wave 9; January 2017- June 2019) with loneliness data of the COVID-19 Social Study (March-May 2020), and found that loneliness levels were higher in the COVID-19 Social Study. In the longitudinal study by Killgore et al. [22], using the US Mechanical Turk (MTurk) crowdsourcing platform, loneliness increased significantly from March to June 2020, particularly from April to May 2020. Only one longitudinal study among the adult general population was identified using pre-outbreak data on loneliness of the study sample. Luchetti et al. [23] assessed trajectories of loneliness among a nationwide sample of American adults in response to COVID-19, using data of assessments in January/ early February 2020 (before the outbreak), late March (during the President's initial "15 Days to Slow the Spread" campaign), and late April (during the "stay-at-home" policies of most states). In this study [23] no significant changes in loneliness were found among the total study sample, in contrast to an increase in social support.

In addition, current population wide studies treated loneliness as a unidimensional construct reflected by sum scores of items. The study by Hyland et al. [24] provided support for the presence of four exclusive subtypes of loneliness, e.g. a low loneliness, a social loneliness, a emotional loneliness, and a social loneliness and emotional loneliness subtype. To gain further insight in the effects of this pandemic on loneliness, we therefore formulated the following third research question:

## Mental health problems and loneliness

In line with the results of non-COVID-19 studies, COVID-19 related studies among the general adult population found positive relationships between loneliness and mental health problems. For example, the large cross-sectional study by González-Sanguino et al. [25] among a sample of Spanish adults found that loneliness was significantly correlated with depression, anxiety, and posttraumatic stress symptoms at the end of March 2020. The large cross-sectional study by Li and Wang [26], conducted at the end of April 2020 among adult UK residents, showed that loneliness was positively associated with having COVID-19 related symptoms. In the cross-sectional study by Palgi et al. [27], conducted between March 15 and April 2020, loneliness among a large sample of Israeli was positively related with depression and general anxiety symptoms. The study by Killgore et al. [22] among American adults showed that loneliness was significantly correlated with depression and suicidal ideation at the three time points during the pandemic.

The findings above suggest that individuals who were lonely before or during the pandemic more often suffered from mental health problems during this period (from pre-, peri- to post-outbreak) compared to individuals who were not lonely. However, it is unknown to what

extent mental health problems increased (or decreased) among individuals who were lonely before *and* after the outbreak, among individuals who were lonely before *or* after the outbreak, and individuals who were not lonely in the pre-outbreak period.

### Research questions

Aim of the present prospective study is to gain further insight in the effects of the COVID-19 epidemic on the mental health, the recovery of symptoms, loneliness and the associations between loneliness and mental health problem among the general population. More specifically:

1. To what extent did the post-outbreak prevalence of anxiety and depression symptom levels (June 2020) differ from the pre-outbreak (November 2019) and peri-outbreak (March 2020) prevalence?

2. To what extent did individuals with anxiety and depression symptom levels during the outbreak (March 2020) differ in recovery of symptoms about three months later (June 2020), from the recovery of individuals with pre-outbreak symptoms (November-December 2019) about three months later (March 2020)?

3. To what extent did the post-outbreak prevalence of emotional loneliness, social loneliness, and social and emotional loneliness (June 2020) differ from the pre-outbreak prevalence (October 2019)?

4. To what extent were individuals with pre-outbreak (October 2019) and post-outbreak (June 2020) emotional and/or social loneliness more at risk for pre- (November 2019), peri- (March 2020) and/or post-outbreak (June 2020) anxiety and depression symptoms?

The objective of this prospective study is to answer these research questions using a Dutch population-based probability sample. Respondents participated in surveys in October 2019 (T1), November 2019 (T2), March 2020 (T3), and June 2020 (T4). With respect to anxiety and depression symptoms we focused on the prevalence of moderate-high anxiety and depression symptom levels, and with respect to recovery we made a distinction between the recovery-categories 'remission', 'improved', 'unchanged' and 'worsened'. Concerning loneliness, we focused on the following subtypes of loneliness: 'low loneliness', 'emotional loneliness', 'social loneliness', and 'social and emotional' loneliness.

## Materials and methods

### Procedures and participants

We extracted and merged data from the Longitudinal Internet studies for the Social Sciences (LISS) panel. This panel is administered by CentERdata and the set-up was funded by the Dutch Research Council (NWO). The LISS panel is based on a traditional probability sample drawn from the Dutch population register by Statistics Netherlands [28]. Participants receive an incentive of 15 euros per hour. Panel members who do not have a computer and/or internet access are provided with the necessary equipment at home (further information about the LISS panel, conducted studies since 2007, and open access data see: https://www.dataarchive.lissdata.nl/; in English). Data of the LISS panel (until March 2020) was also used in the COVID-19 study by van der Velden et al. [12].

Data on loneliness were extracted from two surveys: the (longitudinal) Social Integration and Leisure module of the LISS panel, wave October 2019 (T1: $N^{invited}$ = 5,929, response = 84.2%), and the 'Effects of the Outbreak of Covid-19'-survey conducted in June 2020 (T4: $N^{invited}$ =

7,022, response = 79.7%). Data on anxiety and depression symptoms were extracted from three surveys: from the longitudinal Health module of the LISS panel, wave November 2019 (T2: $N^{invited}$ = 5,954, response = 86.4%), the third wave of the Victims in Modern Society survey [11] survey conducted in March 2020 (T3: $N^{invited}$ = 6,568, response = 83.6%), and the above mentioned 'Effects of the Outbreak of Covid-19 survey' conducted in June 2020 (T4). In total 4,084 respondents participated in all four surveys. To optimize the representativeness of the study sample, we weighted the data using 16 exclusive demographic profiles among the total adult Dutch population 2019 ($N^{2019}$ = 13,926,066), based on the data of Statistics Netherlands (see: https://opendata. cbs.nl/#/CBS/en/; in English). The 16 profiles were constructed using the following demographic characteristics at the end of 2019: sex (male, female), age (18–34, 35–49, 50–64, 65 years and older) and marital status (married and unmarried), yielding 2*4*2 = 16 exclusive demographic profiles. All results are based on the total weighted sample.

## Ethical approval and informed consent

According to the Dutch Medical Research Involving Human Subjects Act (WMO) the present study did not require approval from a Medical Ethical Testing Committee (METC). Nevertheless, the longitudinal Social Integration and Leisure module and Health module (as part of the Longitudinal Core Study in LISS, starting in 2007) were evaluated and approved by the Board of Overseers, an Internal Review Board (IRB) until 2014. The VICTIMS survey (starting in 2018) was approved by an Institutional Review Board, consisting of external and internal reviewers of CentERdata. The 'Effects of the Outbreak of Covid-19' survey was conducted by the Institute for Applied Microeconomics of Universität Bonn in Germany, and in Germany no medical ethical approval is required for this type of research. The 'Effects of the Outbreak of COVID-19' survey was evaluated and approved by internal reviewers of CentERdata not involved in designing this study, and used the same questions on mental health and loneliness as the surveys approved by the IRB. In accordance with the General Data Protection Regulation (GDPR), participants gave explicit written consent for the use of the collected data for scientific and policy relevant research.

## Measures

**Anxiety and depression symptoms.** Anxiety and depression symptoms were assessed at T2, T3 and T4 using the Mental Health Index or Inventory (5-item subscale of the MOS 36-item short-form health survey) [29, 30]. The MHI-5 asks respondents to rate their mental health during the past month on 6-point Likert scales, such as 'This past month I felt very anxious' and 'I felt depressed and gloomy' (0 = never, 1 = seldom, 2 = sometimes, 3 = often, 4 = mostly, 5 = continuously). After recoding the three negative formulated items, the total scores were computed and multiplied by four (to arrive at a 0–100 scale) where lower scores indicate more anxiety and depression symptoms levels (all Cronbach's Alpha's $\geq$ 0.87). A cut-off score of < 60 [31] was used to identify respondents with moderate to high anxiety and depression symptom levels.

**Loneliness.** Loneliness was assessed at T1 and T4, using the 6-item De Jong Gierveld Loneliness Scale [32, 33]. Respondents are asked to rate items such as 'I often feel deserted' and 'there are enough people I can count on in case of a misfortune' on three-point Likert scales (1 = yes, 2 = more or less, 3 = no). According to the scoring guidelines the items were dichotomized to reflect the 'presence' or 'absence' of an indicator (item) of loneliness. For the emotional loneliness items, agreement responses were taken to indicate item endorsement, while for the social loneliness items, disagreement responses were taken to indicate item endorsement. Given the analytic strategy (see below) we did not compute Cronbach's Alpha's.

**Physical illness.** Physical illness was assessed at T2. The Health module assessed several Physician-diagnosed Diseases (PD) in the past year (1 = yes, 2 = no) and Health Problems (HP) respondents regularly suffer from (1 = yes, 0 = no). For the present study we focused on reported: 1.) lung problems ((PD = chronic lung disease such as chronic bronchitis or emphysema or asthma) or (HP = short of breath, problems with breathing, or coughing, a stuffy nose or flu-related complaints)); 2.) heart problems ((PD = angina, pain in the chest, a heart attack including infarction or coronary thrombosis or another heart problem including heart failure) or (HP = heart complaints or angina, pain in the chest due to exertion)); and 3.) diabetes (PD = diabetes or a too high blood sugar level).

**Data analyses.** Repeated measures multivariate logistic regression analyses were conducted to answer the first research question, while controlling for sex and pre-outbreak age, education level, employment status, domestic situation, employment status, ethnicity, lung problems, heart problems, and diabetes at the end of 2019 (November-December).

To answer the second research question, we applied the Reliable Change Index (RCI) and clinical cut-off (CO)-score [34] to gain insight in how many respondents with moderate-high symptom levels at T2 recovered (remission), improved, faced similar symptom levels (unchanged) or suffered from higher symptom levels at T3 (worsened). The psychometric data required to compute both RCI and CO was based on the data of the total weighted sample at T2 (general population; N = 5.157). Computation was done as follows:

1. $RCI = 1.96^* \sqrt{2 \, (SEM)^2}$, with $SEM = SD^*(1-Cronbach's\ \alpha)$; and

2. $CO = ((M^{\text{mod.-high symptom subgroup}} * SD^{\text{general population}}) + (M^{\text{gen. population}} * SD^{\text{mod.high symptom subgroup}})) / (SD^{\text{general}} * SD^{\text{mod.high symptom subgroup}})]$.

The prevalence of four categories of recovery (change) at T3 among respondents with moderate-high anxiety and depression scores at T2 were computed as follows [35]:

1. *Remission* $((M^{\text{difference T2-T3}} \geq RCI)$ and $(M^{T2} \geq M^{T3})$ and $(M^{T3} < CO))$;

2. *Improved* $((M^{\text{difference T2-T3}} \geq RCI)$ and $(M^{T2} \geq M^{T3})$ and $(M^{T3} \geq CO))$;

3. *Unchanged* $(M^{\text{difference T2-T3}} < RCI)$; and

4. *Worsened* $((M^{\text{difference T2-T3}} < RCI)$ and $(M^{T2} < M^{T3})$ nd $(M^{T3} \geq CO))$.

The same procedure was used to assess recovery at T4 among those with moderate-high anxiety and depression symptom levels at T3. For these analyses MHI-scores were reversed (100-score), where higher scores reflect more symptoms.

The subgroup of respondents with moderate-high anxiety and depression symptom levels at T3 partly overlapped with the subgroup with high symptom levels at T2. Of the 961 respondents with high symptom levels at T2 *or* T3, 431 had high symptoms levels at T2 *and* T3. Because of this overlap, differences in the distribution of these four recovery categories at T3 and T4 could not directly be assessed using a chi-squared test. To solve this non-independence problem and to be able to compare the distribution of recovery categories at T3 and T4 we used the following analytic strategy to obtain exclusive subsamples. Firstly, we randomly split the total study sample into two almost equal groups of respondents (A and B). This resulted in two independent subgroups with high symptom levels at T2 (subgroup A1 and B1) and two independent subgroups with high symptoms levels at T3 (subgroup A2 and B2). Secondly, we assessed the differences in the distribution of recovery categories between the independent subgroups A1 and B2, and between the independent subgroups B1 and A2 using a chi-squared test.

To answer the third research question, we first performed latent class analyses (LCA) to identify subtypes of loneliness with the dichotomized items of the De Jong Gierveld Loneliness

scale as indicators. The analyses were performed for T1 and T4 separately. Given the results of Hyland et al. [24] that identified four classes using the instrument, we limit the comparisons of one to seven classes by assessing the relative fit based on reductions in the Bayesian information-criterion (BIC), the sample size adjusted BIC, as well as bivariate residuals (BVR's) [36, 37]. The most parsimonious model with the lowest BIC and sample size adjusted BIC was chosen, while examining the corresponding BVR's of the model that seemed to be the most parsimonious (BVR's larger than 3.84 indicate that the estimated models failed to account for the pairwise association of the specific indicators or covariates). In addition, we looked at entropy values that are measures of the quality of classification (approaching one indicates clear delineation of classes) [38]. LCA was conducted using LatentGold 5.1. Subsequently, class membership was added to the respondents' data. Possible changes in loneliness, e.g. changes in the prevalence of identified classes, were assessed using repeated measures multivariate logistic regression while controlling for the same variables. All non-LCA analyses were conducted with IBM SPSS version 26.

For the fourth research question, multivariate logistic regression analyses and repeated measures multivariate logistic regression were performed with pre-, peri- and post-outbreak anxiety and depression symptoms as dependent variables, while controlling for the same variables as in the previous analyses. Similar to Hyland et al. [24] the LCA revealed four classes at T1 and T4. The four pre-outbreak (T1) and four post-outbreak (T4) classes were used to construct four exclusive subgroups consisting of respondents who were:

1. *not lonely at T1 and not lonely at T4*;

2. *not lonely at T1, but social and/or emotional lonely* at T4;

3. *social and/or emotional lonely at T1, but not lonely at T4*; and

4. *social and/or emotional lonely at both T1 and T4*.

Although 4*4 subgroups can be distinguished using the four loneliness classes at T1 and T4, we limited the number of subgroups because of low cell counts in several subgroups (see S1 Appendix), but also to improve the power of the statistical tests. Four repeated measures multivariate logistic regression analyses were performed to examine the course of moderate-high anxiety and depression symptom levels (T2 to T4) within each subgroup, while series of multivariate logistic regression were performed to examine differences between those four subgroups in the prevalence of moderate-high anxiety and depression symptom levels at T2, T3 and T4.

## Results

### Characteristics respondents

The characteristics of the respondents are shown in Table 1.

### Changes in prevalence anxiety and depression symptoms

The prevalence of moderate-high anxiety and depression symptom levels was 16.8%, 17.2% and 15.3% at T2, T3 and T4 respectively. Repeated measures multivariate logistic regression analyses showed that the prevalence decreased significantly at T4 compared to T2 (adjusted OR (AOR) = 0.88, 95% CI = 0.79–0.99, p = 0.026), and compared to T3 (AOR = 0.88, 95% CI = 0.81–0.97, p = 0.007), whereas it did not change significantly between T2 and T3 (AOR = 1.02, 95% CI = 0.94–1.12, p = 0.611).

**Table 1. Characteristics study sample (N = 4,084).**

| | November—December 2019 |
|---|---|
| | n (%) |
| Sex | |
| • male | 2,012 (49.3) |
| • female | 2,072 (50.7) |
| Education level | |
| • low | 991 (24.3) |
| medium | 1,474 (36.1) |
| • high | 1,620 (39.7) |
| Domestic situation | |
| • (un)married cohabitation without child(ren) | 1,399 (34.3) |
| • (un)married cohabitation with child(ren) | 1,382 (33.8) |
| • single with child(ren) | 237 (5.8) |
| • single | 947 (23.2) |
| • other | 118 (2.9) |
| Employment status | |
| • paid employment | 2,005 (49.1) |
| • self-employed | 223 (5.5) |
| • unemployed/job seeker | 96 (2.4) |
| • student | 345 (8.4) |
| • takes care of housekeeping | 317 (7.8) |
| • pensioner | 775 (19.0) |
| • has (partial) work disability | 185 (4.5) |
| • other | 138 (3.4) |
| Age categories (in years) | |
| • 18–34 | 1,089 (26.7) |
| • 35–49 | 966 (23.7) |
| • 50–64 | 1,057 (25.9) |
| • 65 and older | 972 (23.8) |
| Ethnicity | |
| • native | 3,203 (78.4) |
| • non-native | 881 (21.6) |
| Lung problems | |
| • no | 3,258 (79.8) |
| • yes | 826 (20.2) |
| Heart problems | |
| • no | 3,825 (93.7) |
| • yes | 259 (6.3) |
| Diabetes | |
| • no | 3,908 (95.7) |
| • yes | 176 (4.3) |

self-employed = autonomous professional, freelancer, self-employed, or works or assists in family business.

Educational level: Low = primary education, preparatory intermediate vocational education, or other,

medium = higher general secondary/pre-university education, intermediate professional education, high = higher professional education/university.

### Recovery anxiety and depression symptoms in months after T2 and T3

In Table 2 the prevalence of the four recovery categories at T3 and T4 of the total study sample and the two split-half samples are presented. A Chi-squared test revealed no significant differences in the distribution of recovery categories at T3 and T4 between subgroup A1 and B2 ($\chi^2$ = 3.726, df = 3, p = 0.293), nor between subgroup A2 and B1 ($\chi^2$ = 2.58, df = 3, p = 0.461) respectively. These analyses were repeated among those with high anxiety and depression symptom levels (using cut-off < 45 of original scale). No differences in recovery were found (see S2 Appendix). Because of the random sampling to obtain the two half-split samples (see Method section), we repeated the random sampling to rule out the possibility that findings can be attributed to one specific sampling. These control analyses also showed no differences.

### Loneliness subtypes

The results of the LCA showed, similar to Hyland et al. (2019), that the 4-class solution offered the best parsimonious fit with the lowest BIC and SABIC values, relatively low classification errors ($\leq$ 5%) and high Entropy R's ($\geq$ 0.91) at both T1 and T4 (see S3 Appendix for all details). Given the probabilities of endorsing loneliness items on the four identified classes at T1 and T4 (see S4 Appendix for all details), in parallel with Hyland et al. (2019), class 1 was labelled "low loneliness", class 2 "emotional loneliness", class 3 "social and emotional loneliness" and class 4 was labelled "social loneliness". The distribution of classes at T1 and T4 is presented in Table 3 and indicates that the large majority was not social and/or emotional lonely at T1 or T4. The prevalence of anxiety and depression symptoms at T2 among the low loneliness, emotional loneliness, social and emotional loneliness, and social loneliness class at T1 (one month before T2) differed significantly ($\chi^2$ = 612, 29, df = 3, p < .001). The prevalence was 8.9%, 40.5%, 54.2% and 25.4% respectively. The same pattern was found at T4 ($\chi^2$ = 786.32, df = 3, p < .001): the prevalence was 5.5%, 36.3%, 60.6%, and 23.9% respectively.

**Table 2. Symptoms, recovery, loneliness before and during the COVID-19 among general population.**

| | Anxiety and depression symptoms at T3/T4 | | | | | |
|---|---|---|---|---|---|---|
| | | | split-half groups[1] | | | |
| | Total group | | Subgroup A1 | | Subgroup B1 | |
| | n | % | n | % | N | % |
| Anxiety and depression symptoms at T2 | | | | | | |
| • remission at T3 | 110 | 16.1 | 56 | 17.0 | 58 | 16.2 |
| • improved at T3 | 37 | 5.4 | 16 | 4.9 | 17 | 4.8 |
| • unchanged at T3 | 373 | 54.3 | 179 | 54.4 | 194 | 54.3 |
| • worsened at T3 | 166 | 24.2 | 78 | 23.7 | 88 | 24.6 |
| | Total group | | Subgroup A2 | | Subgroup B2 | |
| | n | % | n | % | n | % |
| Anxiety and depression symptoms at T3 | | | | | | |
| • remission at T4 | 127 | 18.0 | 69 | 20.2 | 57 | 15.7 |
| • improved at T4 | 42 | 6.0 | 18 | 5.3 | 25 | 6.9 |
| • unchanged at T4 | 395 | 56.1 | 183 | 53.5 | 213 | 58.5 |
| • worsened at T4 | 140 | 19.9 | 72 | 21.1 | 69 | 19.0 |

[1]The total study sample (N = 4084) was randomly split in two exclusive subgroups (subgroup A = 2038 and subgroup B = 2046). Due to weighting the distribution is not exactly 50–50% and therefore the numbers of the subgroups sometimes differ from the total numbers. For all four subgroups recovery was computed like the computations of recovery at T3 and T4 among the total group with moderate-high symptom levels at T2 and T3, using the M's and SD's of the four subgroups.

**Table 3. Course of loneliness classes.**

| | October 2019 (T1) | | June 2020 (T4) | | |
|---|---|---|---|---|---|
| Loneliness | n | % | n | % | AOR (95% CI) |
| • low loneliness (class 1) | 3,095 | 75.8 | 2,862 | 70.1 | 0.68 (0.63–0.74)*** |
| • emotional loneliness (class 2) | 753 | 18.4 | 1014 | 24.8 | 1.61 (1.48–1.76)*** |
| • social and emotional loneliness (class 3) | 168 | 4.1 | 142 | 3.5 | 0.84 (0.69–1.01) |
| • social loneliness (class 4) | 68 | 1.7 | 67 | 1.6 | 0.97 (0.72–1.30)[1] |

AOR = Odd Ratios adjusted for age, sex, education, employment status, domestic situation, employment status, ethnicity, lung problems, heart problems, diabetes. 95% CI = 95% Confidence interval. Due to weighting, numbers may slightly differ between Tables.

*** $p < .001$.

[1] AOR without employment status as covariate because of computation problems due to low cell counts.

## Changes in loneliness

Repeated measures multivariate logistic regression analyses (see Table 3) showed that the prevalence of respondents with low loneliness decreased significantly between T1 and T4 (75.8% and 70.1% respectively), while the prevalence of emotional loneliness increased in this period (18.4% and 24.4% respectively). The prevalence of 'social and emotional loneliness' and 'social loneliness' did not differ significantly between T1 and T4.

## The associations between changes in loneliness, and anxiety and depression symptoms

Table 4 provides an overview of the prevalence of moderate-high anxiety and depression symptom levels at T2, T3 and T4 of the subgroups with and without changes in loneliness [$N^{\text{not lonely at T1, not lonely at T4}}$ = 2559 (62.7%); $N^{\text{not lonely at T1, lonely at T4}}$ = 536 (13.1%); $N^{\text{lonely at T1, not lonely T4}}$ = 303 (7.4%); $N^{\text{lonely at T1, lonely at T4}}$ = 686 (16.8%)]. The results of the repeated measures multivariate logistic regression analyses show no changes in the prevalence of anxiety and depression symptoms between T2 and T3 across the four subgroups. However, in the 'not lonely at T1, not lonely T4 subgroup' and 'lonely at T1, not lonely at T4' subgroup, the prevalence of anxiety and depression symptoms decreased significantly to 4.4% and 14.5%

**Table 4. Course of loneliness and anxiety and depression symptoms.**

| | | | Anxiety and depression symptoms | | | | | | | T2 vs. T3 | T2 vs. T4 | T2 vs. T4 |
|---|---|---|---|---|---|---|---|---|---|---|---|---|
| | Total | | November 2019 | | March 2020 | | June 2020 | | | | | |
| | | | (T2) | | (T3) | | (T4) | | | | | |
| Loneliness | n | % | n | % | n | % | n | % | | AOR (95% CI) | AOR (95% CI) | AOR (95% CI) |
| not lonely at T1, not lonely at T4 | 2559 | 62.7 | 178 | 7.0[a] | 205 | 8.0[a] | 112 | 4.4[a] | | 1.17 (0.99–1.39) | 0.62 (0.49–0.77)*** | 0.52 (0.43–0.64)*** |
| not lonely at T1, lonely at T4 | 536 | 13.1 | 96 | 17.9[b] | 104 | 19.4[b] | 141 | 26.3[b] | | 1.10 (0.86–1.39) | 1.63 (1.24–2.14)*** | 1.49 (1.18–1.87)*** |
| lonely at T1, not lonely T4 | 303 | 7.4 | 65 | 21.5[b] | 64 | 21.1[b] | 44 | 14.5[c] | | 0.91 (0.67–1.24) | 0.60 (0.41–0.90)* | 0.66 (0.47–0.92)* |
| lonely at T1, lonely at T4 | 686 | 16.8 | 348 | 50.7[c] | 332 | 48.4[c] | 329 | 48.0[d] | | 0.88 (0.75–1.04) | 0.88 (0.72–1.07) | 0.99 (0.84–1.17) |

AOR = Odd Ratios adjusted for age, sex, education, employment status, domestic situation, employment status, ethnicity, lung problems, heart problems, diabetes.

Percentages in a column with the same letter in superscript ([a, b, c, d]) do not differ significantly according to adjusted Odd Ratios. 95% CI = 95% Confidence interval.

Due to weighting, numbers may slightly differ between Tables.

* $p < 0.05$,

** $p < 0.01$,

*** $p < .001$. Lonely = social loneliness, emotional loneliness, or social and emotional loneliness.

respectively. In subgroup 'not lonely at T1, lonely at T4' the prevalence rose significantly from 17.9% at T1 to 26.3% at T4.

Despite the absence of changes in the prevalence of anxiety and depression symptoms between T2 and T3 across the four subgroups based on the loneliness classes at T1 and T4, the multivariate logistic regression clearly show significant differences in the prevalence between the four groups at T2, T3, and T4. For example, at T4, the prevalence of moderate-high anxiety and depression symptoms levels differed significantly between all subgroups (4.4%, 26.3%, 14.5% and 48% respectively). As expected, respondents who were not lonely at pre- and post-outbreak had the lowest prevalence of moderate-high anxiety and depression symptom levels at T2, T3 and T4.

## Discussion

To date, the present study on the effects of COVID-19 on mental health and loneliness among the general population is one of the very few prospective studies based on probability-based samples of the general population, using data on mental health and loneliness of the same study sample just before the COVID-19 outbreak (October-November 2019). The results showed the following clear patterns. Compared to the pre-outbreak (T2) and outbreak (T3) prevalence of anxiety and depression symptoms, the prevalence of post-outbreak (T4) symptoms did not increase but decreased slightly but significantly. Interestingly, the study by Wang et al. [39] also showed a significant reduction in depression and anxiety symptoms during the stressful first four weeks after the COVID-19 outbreak in China.

These results seem to differ strongly from the studies by McGinty et al. [22] and Daly et al. [9], and especially Twenge and Joiner [7], but we cannot rule out the possibility that differences in study designs, among other things, play a role in these differences. For instance, these studies compared different pre- and post-COVID outbreak study samples instead of pre- and post-COVID-19 outbreak assessments within the same sample. However, the study by Pierce et al. [11], similar to our study, prospectively assessed mental distress among the same sample showing an increase from 2018–2019 to April 2020. On the other hand, in their population-based study Hyland et al. [40] concluded that the rates of screening positive for generalized anxiety disorder and depression in Ireland around the beginning of April 2020, did not differ markedly from those reported in previous non-COVID-19 national prevalence studies in the UK.

As described earlier [12] the absence of differences in anxiety and depression symptoms within the Dutch study sample may also be related to how the Dutch governments reacted and intervened, and related to the Dutch social welfare and health care systems in the Netherlands. Unemployed adults (or adults who lost their job due to this pandemic) can invoke for unemployment benefits and, in principle, each Dutch citizen has a health care insurance regardless of being employed or not. During the study period the Dutch government very quickly implemented financial support to companies who lost revenues significantly because of the COVID-19 pandemic to allow them to keep people at work. The governmental acts, interventions and management of this pandemic were, until now, not politized in the Netherlands, meaning that the compensation for lost salaries and profits was rather generous, preventing potential sources of additional financial worries and stress to the people concerned.

A relevant research topic not covered in previous longitudinal COVID-19 studies is the recovery of existing mental health problems during this pandemic [14]. We examined, using the Reliable Change Index (RCI) of Jacobson and Truax [34] the extent to which individuals with anxiety and depression symptoms at the beginning of the COVID-19 outbreak (T3) recovered a few months later (T4). We compared the distribution of the four recovery-categories (remission, improved, unchanged and worsened) with the distribution of these categories during the outbreak (T3) among those with anxiety and depression symptoms before the

outbreak (T2) and found no significant differences. Moreover, additional analyses among individuals with very high existing MHI-5 scores showed similar results. In other words, we found no support in the data that the COVID-19 pandemic hindered recovery of anxiety and depression symptoms during the first four months. This suggest that we have to be careful to attribute worsening symptoms solely to this pandemic. In addition, our findings do not deny or neglect that people with existing anxiety and depression symptoms may have concerns about their mental illness worsening because of the COVID-19 pandemic [41].

In line with the study of Hyland et al. [24], the latent class analyses revealed four exclusive classes of loneliness: the low loneliness, emotional loneliness, social loneliness, and emotional and social loneliness class. Findings with respect to the course of loneliness during the pandemic differed partly from the findings with respect to anxiety and depression symptoms. The analyses showed that the prevalence of emotional loneliness increased after the outbreak (18.4% versus 24.8%), while the prevalence of individuals who were social lonely or social and emotional lonely did not differ from the pre-outbreak prevalence. The prevalence of individuals with low loneliness decreased slightly from 75.8% to 70.1%, suggesting that the lockdowns and social distancing (including stay-at-home orders) negatively affected loneliness for a relatively small group [9, 11, 22]. The absence of differences between pre- and post-outbreak social loneliness and social and emotional loneliness suggest that people have found ways to keep social contacts, such as via (mobile) telephone, social media (e.g. Whatsapp, Facebook) and video calls (e.g. Skype, ZOOM), but that these communication facilities may not always provide emotional connectedness. Although we controlled for demographics and physical illness, it was outside the aim and scope of the present study to further discuss which groups were more at risk to become emotional lonely. Our results seem to differ from the outcomes of the prospective study of Luchetti et al. [23], the only other prospective study on loneliness among the general population using a longitudinal sample. In their study, however, loneliness was treated as a unidimensional concept.

As expected, and in line with previous research, individuals with persistent loneliness (lonely before and after the COVID-19 outbreak) much more often suffered from anxiety and depression symptoms before, during and after the outbreak (prevalence $\geq$ 48%) than individuals who were not lonely before and after the outbreak (prevalence $\leq$ 8%). Importantly, results furthermore showed that the prevalence of symptoms among the four subgroups (not lonely at T1, not lonely at T4; not lonely at T1, lonely at T4; lonely at T1, not lonely at T4; lonely at T1, lonely at T4) did not change between November 2019 and March 2020 suggesting that the effects of all social distancing measures took place after the outbreak. Interestingly, among those who became lonely, the prevalence of symptoms increased, while among those who were not lonely anymore after the outbreak or not lonely before and after the outbreak, the prevalence of symptoms decreased after the outbreak (June 2020). These findings indicate that loneliness plays a key role in the presence of anxiety and depression symptoms during this pandemic. The results also suggest that COVID-19 related mental health policies should target individuals who became lonely after the COVID-19 outbreak (13.1% of the total study sample) and especially target individuals who were and are lonely after the outbreak (16.8% of the total study sample). Unfortunately, due to the low cell counts we could not examine the prevalence of symptoms among all sixteen combinations of the pre- and post-outbreak loneliness classes. Future studies are warranted to examine loneliness on the longer term to assess the extent to which patterns in loneliness remain stable or change.

## Strengths and limitations

This study has its strengths and limitations. The prospective study design with pre-COVID-19 data on loneliness, anxiety and depression symptoms, use of a longitudinal population-based

probability sample with high response-rates, and use of well-validated questionnaires are major strengths of the current study. We used the data of a large group of respondents (N = 4,084) who participated in all four waves. However, we did not conduct clinical interviews to assess mental disorders such as generalized anxiety disorder and schizophrenia nor examined other relevant mental health problems such as sleeping problems and fatigue, which would have enriched our study. For instance, we cannot rule out the possibility that the recovery of depression, fatigue or sleeping problems after the COVID-19 outbreak is, due to all (practical) changes, lower than before the outbreak. In addition, we were not able to assess mental health consequences among people specific mental disorders such as schizophrenia [42]. The most recent assessment in this study took place in June 2020 and future follow-ups are needed to gain insight in the effects of the pandemic on the longer term given the uncertainties about how this pandemic will develop and when effective vaccines will become available. In addition, we did not include children and adolescents of the Dutch general population who might be affected differently by the COVID-19 pandemic. We have no data on anxiety and depression symptoms, and loneliness covering an exact similar study period (October-November to June) to rule out possible seasonal effects: it is basically possible that for instance the decrease of the prevalence of anxiety and depression symptoms is caused by this effect.

Results in this paper are based on data that were representative for the Dutch population. It is unclear to what extent the results can be generalized to other developed countries who were hit harder by the COVID-19 pandemic or where the individual-level employment consequences were more severe, let alone the global South. The Dutch economy is projected to recover quickly from the COVID-19 health crisis in 2021 with a reduction in economic growth of -4% in 2020 and a positive forecasted growth of almost 3% in 2021 [43], and countries differ on these topics [44].

## Final conclusions

Taken together, these results suggest that the Dutch population was capable to cope with and to adjust successfully to the drastic changes of their lives as a result of the COVID-19 pandemic outbreak during the first four months of 2020 (March 2020 to June 2020; cf. [12, 45, 46]). We did not find an increase in the prevalence of anxiety and depression symptoms during the first four months of the outbreak. And among those with symptoms we found similar recovery-rates during the first four months compared to before the pandemic. Nevertheless, the prevalence of emotional loneliness increased to some extent, and an increase in loneliness, especially persistent loneliness, was very strongly associated with the presence of anxiety and depression symptoms.

About 100 years ago (1918–1919) the world was confronted with another global devastating pandemic; the Spanish flu. It is estimated that about one-third of the world's population was infected and at least 50 million people were killed by the Spanish flu [47]. To gain further insight in the effects of the COVID-19 pandemic it might be of interest, besides future empirical research following the COVID-19 outbreak, to include historical research on the effects of the Spanish flu [48] on for instance (mental) health, family life and local communities. It may further help to formulate future research questions based on the lessons learned from the Spanish flu.

## Supporting information

**S1 Appendix. Tabel S1.** Cross-tabulation loneliness classes at T1 and T4.
(DOCX)

**S2 Appendix. Table S2.** Recovery in the months following at T2 and T3.
(DOCX)

**S3 Appendix. Table S3.** Results LCA loneliness.
(DOCX)

**S4 Appendix. Table S4.** Latent class profiles of loneliness at T1 and T4.
(DOCX)

## Author Contributions

**Conceptualization:** Peter G. van der Velden, Philip Hyland, Carlo Contino, Hans-Martin von Gaudecker, Ruud Muffels, Marcel Das.

**Data curation:** Peter G. van der Velden, Hans-Martin von Gaudecker.

**Formal analysis:** Peter G. van der Velden, Marcel Das.

**Funding acquisition:** Peter G. van der Velden, Marcel Das.

**Investigation:** Peter G. van der Velden, Philip Hyland, Carlo Contino, Hans-Martin von Gaudecker, Ruud Muffels, Marcel Das.

**Methodology:** Peter G. van der Velden, Philip Hyland, Hans-Martin von Gaudecker, Ruud Muffels, Marcel Das.

**Project administration:** Peter G. van der Velden.

**Resources:** Peter G. van der Velden.

**Supervision:** Peter G. van der Velden.

**Validation:** Peter G. van der Velden, Philip Hyland, Carlo Contino, Hans-Martin von Gaudecker, Ruud Muffels, Marcel Das.

**Writing – original draft:** Peter G. van der Velden.

**Writing – review & editing:** Peter G. van der Velden, Philip Hyland, Carlo Contino, Hans-Martin von Gaudecker, Ruud Muffels, Marcel Das.

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
