## [Decision Letter · Decision Letter 0]

4 Dec 2020

PONE-D-20-31580

Anxiety and depression symptoms, the recovery from symptoms and loneliness before and after the COVID-19 outbreak among the general population. Findings from a Dutch population-based longitudinal study

PLOS ONE

Dear Dr. van der Velden,

Thank you for submitting your manuscript to PLOS ONE. After careful consideration, we feel that it has merit but does not fully meet PLOS ONE’s publication criteria as it currently stands. Therefore, we invite you to submit a revised version of the manuscript that addresses the points raised during the review process.

Both reviewers were positive about your work and believe with some minor revisions, it could be suitable for publication. I agree with their assessment. Please see their detailed and useful comments and  be sure to address each in your revision.

We look forward to receiving your revised manuscript.

Kind regards,

Fuschia M. Sirois, PhD

Academic Editor

PLOS ONE

Journal Requirements:

3.We note that you have stated that you will provide repository information for your data at acceptance. Should your manuscript be accepted for publication, we will hold it until you provide the relevant accession numbers or DOIs necessary to access your data. If you wish to make changes to your Data Availability statement, please describe these changes in your cover letter and we will update your Data Availability statement to reflect the information you provide.

Reviewers' comments:

Reviewer's Responses to Questions

**Comments to the Author**

1. Is the manuscript technically sound, and do the data support the conclusions?

Reviewer #1: Yes

Reviewer #2: Yes

2. Has the statistical analysis been performed appropriately and rigorously? 

Reviewer #1: Yes

Reviewer #2: Yes

3. Have the authors made all data underlying the findings in their manuscript fully available?

Reviewer #1: Yes

Reviewer #2: Yes

4. Is the manuscript presented in an intelligible fashion and written in standard English?

Reviewer #1: Yes

Reviewer #2: Yes

5. Review Comments to the Author

Reviewer #1: This research article aims to gain an insight into the effects of the Covid-19 pandemic on mental health and loneliness. In the current paper, the authors have extracted data from a sample of the Dutch population and examined loneliness and mental health using well-established validated measures. This is a timely piece of research conducted during the pandemic, and one of the few prospective studies I have seen that have studied the impact of covid-19 on mental health and loneliness, and so I welcome this piece of research to the field. I do have a few minor comments.

1. In the abstract it would be useful to have some p values alongside percentages if possible.

2. The 4 research questions are scattered throughout the introduction and I wonder if these could be either summarised at the end of this section or moved to the Methods section.

3. The authors may wish to make reference to whether these results can be generalised (perhaps in the limitations section).

4. Minor suggestions for re-wording and few errors on the attached pdf.

Reviewer #2: This is a really interesting and well conducted study on an extremely important topic, and the results are important, both now and for the future. I don't have any major suggestions for improvements because the research questions you selected and the methods you used to investigate them were sound, the analysis was detailed and the presentation of results and discussion very carefully done.

One area that interested me was in the interpretation of results. Given your method, one element that was naturally missing was the self-reported in depth perspectives of participants, and a qualitative component would have been needed to pursue that. I appreciate this was not what you set out to do, and your approach is absolutely valid. However, I was left wondering, as perhaps you were, how to interpret some of the results. Given how multi-factorial mental health and loneliness are, and how unusual the global pandemic has been, were there were some aspects of the common experience which might have impacted on individuals' experience? For example, did the seriousness of the situation, and our common experience across the country and indeed around the world, provide a unifying experience for people, who may otherwise not have felt that they had much in common with others, and thus counteract the tendency for lockdown to engender loneliness? What has really been going on when so many of us have been connecting through online meetings and video calls - how has this effected levels of isolation? Did the amount of expressed emotion during 2020, which we know has connections to psychological well-being, have an impact on anxiety and depression as we faced the pandemic and discussed it widely? Could the experience of existential or ontological anxiety which many of us have faced have a deeper impact on anxiety levels, including paradoxically to decrease everyday or ontic anxiety? I mention these ideas not to criticise your work, which is excellent and very thought-provoking, but to suggest that there could be other processes which your study was not unreasonably not designed to detect, but which could have had a protective effect on participants and showed up in the data.

I don't think you necessarily need to add anything to your discussion, which was a very good interpretation of your results. These thoughts just occurred to me, and I therefore wanted to pass them on for interest.

Finally, there was a typo error, I believe, on page 21. You state:

"Firstly, we randomly split the total study sample into two almost equal groups of respondents (A and B), This resulted in two

independent subgroups with high symptom levels at T2 (subgroup A1 and B1) and two independent subgroups with high symptoms levels at T3 (subgroup A1 and B2)."

I believe this should have been (subgroup A2 and B2)...?

Congratulations on producing an excellent paper :-)

6. PLOS authors have the option to publish the peer review history of their article (what does this mean?). If published, this will include your full peer review and any attached files.

Reviewer #1: No

Reviewer #2: **Yes: **Chris Blackmore

---

## [Author Response · Author response to Decision Letter 0]

9 Dec 2020

We would like to thank the reviewers very much for their time and effort, and very clear suggestions to improve our manuscript. We really appreciated the kind words and compliments of the reviewers.

REVIEWER #1: 

1. This research article aims to gain an insight into the effects of the Covid-19 pandemic on mental health and loneliness. In the current paper, the authors have extracted data from a sample of the Dutch population and examined loneliness and mental health using well-established validated measures. This is a timely piece of research conducted during the pandemic, and one of the few prospective studies I have seen that have studied the impact of covid-19 on mental health and loneliness, and so I welcome this piece of research to the field. I do have a few minor comments.

Response:

Thank you very much for your kind words.

2. In the abstract it would be useful to have some p values alongside percentages if possible.

Response:

Based on this comment we added the p-values, but after adding the p-values we crossed the word-count limit of 300 words (even after we tried to shorten other sections). However, we agree with the reviewer that information about statistical significance was missing, and we therefore revised the results sections as follows (and shortened the Objectives part of the Abstract):

“Results. Repeated measures multivariate logistic regression analyses (RMMLRA) showed a statistical significant lower prevalence of anxiety and depression symptoms after the outbreak (T4=15.3%) than before (T2=16.8%) and during the COVID-19 outbreak (T3=17.2%). According to the Reliable Change Index, the distribution of recovery categories (remission, improvement, unchanged and worsening symptoms) after the outbreak did not differ significantly from the distribution of these categories before the outbreak. RMMLRA revealed that the prevalence of emotional loneliness increased significantly after the outbreak (T1=18.4%, T4=24.8%). Among individuals who were not lonely before and after the outbreak the prevalence of symptoms decreased significantly (T2=7.0%, T4=4.4%) and, likewise, among those who were not lonely anymore after the outbreak (T2=21.5%, T4=14.5%). However, the prevalence of symptoms increased significantly among those who became lonely during the pandemic (T2=17.9%, T4=26.3%).”

3. The 4 research questions are scattered throughout the introduction and I wonder if these could be either summarised at the end of this section or moved to the Methods section.

Response:

We agree with the reviewer and have summarized the research questions at the end of the introduction section. The intro sections now ends with:

Aim of the present prospective study is to gain further insight in the effects of the COVID-19 epidemic on the mental health, the recovery of symptoms, loneliness and the associations between loneliness and mental health problem among the general population. More specifically: 

1. To what extent did the post-outbreak ………”

Because we moved the research questions, we added sub-headings in the introduction section to improve the readability of the introduction (new sub-headings are Mental health problems, Recovery of symptoms, Loneliness, Mental health problems and loneliness, Research questions).

4. The authors may wish to make reference to whether these results can be generalised (perhaps in the limitations section).

Response:

Based on this comment, in the limitations section we added:

“Results in this paper are based on data that were representative for the Dutch population. It is unclear to what extent the results can be generalized to other developed countries who were hit harder by the COVID-19 pandemic or where the individual-level employment consequences were more severe, let alone the global South. The Dutch economy is projected to recover quickly from the COVID-19 health crisis in 2021 with a reduction in economic growth of -4% in 2020 and a positive forecasted growth of almost 3% in 2021 [43], and countries differ on these topics [44]..”.

[43] CPB (Centraal Planbureau). Novemberraming: Economische vooruitzichten 2021[In Dutch: November estimates: Economic Forecasts]. Centraal Planbureau (CPB), The Hague, The Netherlands, 2020.

[44] OECD (Organisation for Economic Co-operation and Development). OECD Economic outlook December 2020. Available from: http://www.oecd.org/economic-outlook/december-2020/. Accessed 8-12-2020.

5. Minor suggestions for re-wording and few errors on the attached pdf.

Response:

Thank you very much for your detailed comments. 

5.1. Lots of percentages given here, could some p-values be added from the results section?

Response:

See our response to point 2

5.2. …may benefit from adding a reference for what COVID-19 is (e.g. WHO 2020 reference/report) to give context of the disease and symptoms

Response:

We have revised the first sentence in (and re-numbered all references)

“The COVID-19 pandemic [1] negatively…….

[1] Pascarella G, Strumia A, Piliego C, Bruno F, Del Buono R, Costa F, Scarlata S, Agrò FE. COVID-19 diagnosis and management: a comprehensive review. J Intern Med. 2020; 288:192-206. https://doi.org/10.1111/joim.13091

5.3. re-word sentence perhaps?

Response:

We have deleted this sentence.

5.4. "male" and "female" might work better here

Response:

We have replaced “man” and “women” with “male” and “female”.

5.5. delete extra comma

Response:

We have deleted the comma

5.6. reword - "unemployed adults"

Response:

We have revised this sentence into:

“Unemployed adults (or adults who lost their job due to this pandemic) can invoke for…… “

5.7. "may" not?

Response:

We added “may”.

 

REVIEWER #2: 

6. This is a really interesting and well conducted study on an extremely important topic, and the results are important, both now and for the future. I don't have any major suggestions for improvements because the research questions you selected and the methods you used to investigate them were sound, the analysis was detailed and the presentation of results and discussion very carefully done.

Response:

Thank you very much for your compliments.

7. One area that interested me was in the interpretation of results. Given your method, one element that was naturally missing was the self-reported in depth perspectives of participants, and a qualitative component would have been needed to pursue that. I appreciate this was not what you set out to do, and your approach is absolutely valid. However, I was left wondering, as perhaps you were, how to interpret some of the results. Given how multi-factorial mental health and loneliness are, and how unusual the global pandemic has been, were there were some aspects of the common experience which might have impacted on individuals' experience? For example, did the seriousness of the situation, and our common experience across the country and indeed around the world, provide a unifying experience for people, who may otherwise not have felt that they had much in common with others, and thus counteract the tendency for lockdown to engender loneliness? 

What has really been going on when so many of us have been connecting through online meetings and video calls - how has this effected levels of isolation? Did the amount of expressed emotion during 2020, which we know has connections to psychological well-being, have an impact on anxiety and depression as we faced the pandemic and discussed it widely? Could the experience of existential or ontological anxiety which many of us have faced have a deeper impact on anxiety levels, including paradoxically to decrease everyday or ontic anxiety? I mention these ideas not to criticise your work, which is excellent and very thought-provoking, but to suggest that there could be other processes which your study was not unreasonably not designed to detect, but which could have had a protective effect on participants and showed up in the data.

I don't think you necessarily need to add anything to your discussion, which was a very good interpretation of your results. These thoughts just occurred to me, and I therefore wanted to pass them on for interest.

Response:

Thank you for sharing your thoughts on these issues. Your thoughts automatically reminded us of of Spanish flue 100 years ago and how people dealt with all consequences (estimates of global death rates are between 50 and 100 million): what can we learn from this previous devastating pandemic in this perspective? We think historical research on effects of the Spanish flu of interest for empirical researchers too. It may help to focus and formulate history-informed research questions and hypotheses. Based on your thoughts we therefore added at the end of the discussion section:

About 100 years ago (1918-1919) the world was confronted with another global devastating pandemic; the Spanish flu. It is estimated that about one-third of the world’s population was infected and at least 50 million people were killed by the Spanish flu [47]. To gain further insight in the effects of the COVID-19 pandemic it might be of interest, besides future empirical research following the COVID-19 outbreak, to include historical research on the effects of the Spanish flu [48] on for instance (mental) health, family life and local communities. It may further help to formulate future research questions based on the lessons learned from the Spanish flu.” 

[47] CDC. 1918 Pandemic (H1N1 virus). March 20 2019. Available from: https://www.cdc.gov/flu/pandemic-resources/1918-pandemic-h1n1.html. Accessed 7-12-2020

[48] Spinney L. Pale rider: The Spanish flu of 1918 and how it changed the world. London: Jonathan Cape; 2017.

8. Finally, there was a typo error, I believe, on page 21. You state:

"Firstly, we randomly split the total study sample into two almost equal groups of respondents (A and B), This resulted in two independent subgroups with high symptom levels at T2 (subgroup A1 and B1) and two independent subgroups with high symptoms levels at T3 (subgroup A1 and B2)." I believe this should have been (subgroup A2 and B2)...? 

Response:

The reviewer is correct, and we have revised the sentence into:

This resulted in two independent subgroups with high symptom levels at T2 (subgroup A1 and B1) and two independent subgroups with high symptoms levels at T3 (subgroup A2 and B2).

9. Congratulations on producing an excellent paper :-)

Response:

Thanks a lot. 

Comments of PLOS ONE 

Response:

We have checked and revised our manuscript to meet PLOS ONE's style requirements.

Response:

In the original manuscript we wrote: “In accordance with the General Data Protection Regulation (GDPR), participants gave explicit written consent for the use of the collected data for scientific and policy relevant research.”

We have revised this sentence in:

“In accordance with the General Data Protection Regulation (GDPR), participants gave explicit written consent for the use of the collected data for scientific and policy relevant research”.

Response:

We are not sure if we understand this comment correctly because all requested information was given (“…….Further information about all conducted surveys and regulations for free access to the data, as well as the data files can be found at www.lissdata.nl (in English)….. All data of studies conducted with the LISS panel are anonymized. Data of surveys conducted in 2020 will be added to the open access data archive soon”.

Response

We have included the requested captions.

---

## [Editor Report · Decision Letter 1]

22 Dec 2020

Anxiety and depression symptoms, the recovery from symptoms and loneliness before and after the COVID-19 outbreak among the general population. Findings from a Dutch population-based longitudinal study

PONE-D-20-31580R1

Dear Dr. van der Velden,

We’re pleased to inform you that your manuscript has been judged scientifically suitable for publication and will be formally accepted for publication once it meets all outstanding technical requirements.

Kind regards,

Fuschia M. Sirois, PhD

Academic Editor

PLOS ONE

---

## [Editor Report · Acceptance letter]

28 Dec 2020

PONE-D-20-31580R1 

Anxiety and depression symptoms, the recovery from symptoms, and loneliness before and after the COVID-19 outbreak among the general population. Findings from a Dutch population-based longitudinal study 

Dear Dr. van der Velden:

I'm pleased to inform you that your manuscript has been deemed suitable for publication in PLOS ONE. Congratulations! Your manuscript is now with our production department. 

Kind regards, 

on behalf of

Dr. Fuschia M. Sirois 

Academic Editor

PLOS ONE